# Therapeutic potential of IL6R blockade for the treatment of sepsis and sepsis-related death: A Mendelian randomisation study

Fergus W. Hamilton[1,2]*, Matt Thomas[3], David Arnold[4], Tom Palmer[1], Ed Moran[2], Alexander J. Mentzer[5], Nick Maskell[4], Kenneth Baillie[6], Charlotte Summers[7], Aroon Hingorani[8,9,10], Alasdair MacGowan[2], Golam M. Khandaker[1], Ruth Mitchell[1], George Davey Smith[1], Peter Ghazal[11], Nicholas J. Timpson[1]

1 MRC Integrative Epidemiology Unit, University of Bristol, Bristol, United Kingdom, 2 Infection Science, North Bristol NHS Trust, Bristol, United Kingdom, 3 Intensive Care Unit, North Bristol NHS Trust, Bristol, United Kingdom, 4 Academic Respiratory Unit, University of Bristol, Bristol, United Kingdom, 5 Wellcome Centre For Human Genetics, University of Oxford, Oxford, United Kingdom, 6 Roslin Institute, University of Edinburgh, Edinburgh, United Kingdom, 7 Department of Medicine, University of Cambridge, Cambridge, United Kingdom, 8 UCL Institute for Cardiovascular Science, University College London, London, United Kingdom, 9 UCL BHF Research Accelerator, University College London, London, United Kingdom, 10 Health Data Research UK, London, United Kingdom, 11 Project Sepsis, Cardiff University, Cardiff, United Kingdom

* fergus.hamilton@bristol.ac.uk

**Data Availability Statement:** Most data used in this study are publicly available. For ease, curated data (harmonised SNPs and MR results) are

## Abstract

### Background

Sepsis is characterised by dysregulated, life-threatening immune responses, which are thought to be driven by cytokines such as interleukin 6 (IL-6). Genetic variants in *IL6R* known to down-regulate IL-6 signalling are associated with improved Coronavirus Disease 2019 (COVID-19) outcomes, a finding later confirmed in randomised trials of IL-6 receptor antagonists (IL6RAs). We hypothesised that blockade of IL6R could also improve outcomes in sepsis.

### Methods and findings

We performed a Mendelian randomisation (MR) analysis using single nucleotide polymorphisms (SNPs) in and near *IL6R* to evaluate the likely causal effects of IL6R blockade on sepsis (primary outcome), sepsis severity, other infections, and COVID-19 (secondary outcomes). We weighted SNPs by their effect on CRP and combined results across them in inverse variance weighted meta-analysis, proxying the effect of IL6RA. Our outcomes were measured in UK Biobank, FinnGen, the COVID-19 Host Genetics Initiative (HGI), and the GenOSept and GainS consortium. We performed several sensitivity analyses to test assumptions of our methods, including utilising variants around *CRP* and gp130 in a similar analysis.

In the UK Biobank cohort (N = 486,484, including 11,643 with sepsis), IL6R blockade was associated with a decreased risk of our primary outcome, sepsis (odds ratio (OR) = 0.80; 95% confidence interval (CI) 0.66 to 0.96, per unit of natural log-transformed CRP

available at the authors GitHub (https://github.com/gushamilton/il6-sepsis), so all findings can be replicated. Raw data from the FinnGen GWAS are available via the FinnGen website (https://www.finngen.fi/), COVID-HGI GWAS are available via the COVID-HGI website (https://www.covid19hg.org/), and the UK Biobank GWAS performed as part of this study are available at the MRC-IEU Open GWAS repository (https://gwas.mrcieu.ac.uk/). Access to the full summary statistics for the sepsis GWAS performed by the GaINS and GenOSept committee is by application to the relevant committee (https://ukccggains.com/). This research was performed under UK Biobank application 56243. Individual access to UK Biobank can be arranged via the UK Biobank website (https://www.ukbiobank.ac.uk/).

**Funding:** FH's time was funded by the GW4 CAT Doctoral Fellowship scheme (Wellcome Trust, 222894/Z/21/Z, https://www.gw4-cat.ac.uk). AH's time was funded by UCL British Heart Foundation Accelerator (AA/18/6/34223, https://www.bhf.org.uk/), the UCL NIHR Biomedical Research Centre, and the UKRI/NIHR funded Multimorbidity Mechanism and Therapeutics Research Collaborative (MR/V033867/1, both https://www.nihr.ac.uk/). AJM is a NIHR Academic Clinical Lecturer and supported by the Oxford Biomedical Research Centre (BRC, https://www.nihr.ac.uk/). PG's time was funded by the Welsh Government and the EU-ERDF (Ser Cymru Scheme, https://gov.wales/ser-cymru). CS is supported by funding from National Institute for Health and Care Research (NIHR133788, https://www.nihr.ac.uk/).) and UKRI (MR/X005070/1, https://www.ukri.ac.uk/). Her research programme is also supported by the Cambridge NIHR Biomedical Research Centre (BRC-1215-20014, https://www.nihr.ac.uk/), the Wellcome Trust (https://wellcome.org/), and GlaxoSmithKline plc (https://www.gsk.com/). JKB gratefully acknowledges funding support from a Wellcome Trust Senior Research Fellowship (223164/Z/21/Z, https://wellcome.org/), UKRI grants MC_PC_20004, MC_PC_19025, MC_PC_1905, MRN02995X/1, and MC_PC_20029 (https://www.ukri.org/), Sepsis Research (Fiona Elizabeth Agnew Trust, https://sepsisresearch.org.uk/), a BBSRC Institute Strategic Programme Grant to the Roslin Institute (BB/P013732/1, BB/P013759/1, https://www.ukri.org/councils/bbsrc/), and the UK Intensive Care Society (https://ics.ac.uk/). We gratefully acknowledge the support of Baillie Gifford (https://www.bailliegifford.com/) and the Baillie Gifford Science Pandemic Hub at the University of Edinburgh) NJT is a Wellcome Trust Investigator (202802/Z/16/Z) and works within the University of Bristol National Institute for Health

decrease). The size of this effect increased with severity, with larger effects on 28-day sepsis mortality (OR = 0.74; 95% CI 0.47 to 1.15); critical care admission with sepsis (OR = 0.48, 95% CI 0.30 to 0.78) and critical care death with sepsis (OR = 0.37, 95% CI 0.14 to 0.98).

Similar associations were seen with severe respiratory infection: OR for pneumonia in critical care 0.69 (95% CI 0.49 to 0.97) and for sepsis survival in critical care (OR = 0.22; 95% CI 0.04 to 1.31) in the GainS and GenOSept consortium, although this result had a large degree of imprecision. We also confirm the previously reported protective effect of IL6R blockade on severe COVID-19 (OR = 0.69, 95% CI 0.57 to 0.84) in the COVID-19 HGI, which was of similar magnitude to that seen in sepsis. Sensitivity analyses did not alter our primary results. These results are subject to the limitations and assumptions of MR, which in this case reflects interpretation of these SNP effects as causally acting through blockade of IL6R, and reflect lifetime exposure to IL6R blockade, rather than the effect of therapeutic IL6R blockade.

## Conclusions

IL6R blockade is causally associated with reduced incidence of sepsis. Similar but imprecisely estimated results supported a causal effect also on sepsis related mortality and critical care admission with sepsis. These effects are comparable in size to the effect seen in severe COVID-19, where IL-6 receptor antagonists were shown to improve survival. These data suggest that a randomised trial of IL-6 receptor antagonists in sepsis should be considered.

## Author summary

### Why was this study done?

- Inhibition of the cytokine interleukin 6 (IL-6) using drugs such as tocilizumab, which bind to the IL-6 receptor, has been shown to reduce mortality in critically unwell patients with Coronavirus Disease 2019 (COVID-19).

- It is currently unknown whether IL-6 inhibition might have similar benefits in other, severe infections, such as bacterial sepsis.

- Genetic analyses (Mendelian randomisation (MR)) have previously predicted the success of IL-6 inhibition in COVID-19 and other conditions.

### What did the researchers find?

- In a large, UK cohort ($N$ = 485,825, including 11,643 with sepsis), genetic variation acting as a proxy (or natural experiment) for IL6R blockade was associated with a reduced odds of sepsis (odds ratio (OR) 0.80; 95% confidence interval (CI) 0.66 to 0.96) in MR analyses.

Research (NIHR) Biomedical Research Centre (BRC, https://www.nihr.ac.uk). NJT is supported by the Cancer Research UK (CRUK) Integrative Cancer Epidemiology Programme (C18281/A29019, https://www.cancerresearchuk.org/). The views expressed are those of the authors and not necessarily those of the NIHR, the NHS, or the Department of Health and Social Care. GDS, FH, RM, TP work within the MRC Integrative Epidemiology Unit at the University of Bristol, which is supported by the Medical Research Council (MC_UU_00011/1). G.M.K. acknowledges funding support from the Wellcome Trust (Grant No. 201486/Z/16/Z), the Medical Research Council, UK (Grant No. MC_UU_00011/1; Grant No. MR/S037675/1; and Grant No. MR/W014416/1), and the National Institute of Health Research Bristol Biomedical Research Centre, UK (Grant No. NIHR203315). The funders had no role in study design, data collection and analysis, decision to publish, or preparation of the manuscript.

**Competing interests:** I have read the journal's policy and the authors of this manuscript have the following competing interests: GDS is a member of PLOS Medicine's editorial board. GDS reports Scientific Advisory Board Membership for Relation Therapeutics and Insitro.

**Abbreviations:** CHARGE, Cohorts for Heart and Aging Research in Genomic Epidemiology; CI, confidence interval; COVID-19, Coronavirus Disease 2019; CRP, C-reactive protein; GWAS, genome-wide association study; HES, Hospital Episode Statistics; HGI, Host Genetics Initiative; hsCRP, high-sensitivity CRP; IL-6, interleukin 6; IL6RA, IL-6 receptor antagonist; IVW, inverse variance weighting; LRTI, lower respiratory tract infection; MR, Mendelian randomisation; OR, odds ratio; SNP, single nucleotide polymorphism; URTI, upper respiratory tract infection.

- Effects were consistent in secondary cohorts and when using differing definitions of sepsis, with effect sizes generally larger in more severe phenotypes.

- The effect estimates on sepsis were similar in magnitude to those seen in severe COVID-19 (OR 0.69, 95% CI 0.57 to 0.84), where IL-6 blockade is currently recommended.

## What do the findings mean?

- Within the limitations and assumptions of MR, these findings support the consideration of IL-6 inhibition in randomised controlled trials in sepsis.

- More broadly, these findings support the potentially pathological role of IL-6 in severe infection.

## Introduction

Sepsis is a complex physiological and metabolic response to infection, characterised by dysregulation of the immune response and organ dysfunction [1]. Our current best treatments remain antimicrobial therapy and organ support, with no licenced treatments outside these interventions [2].

Interleukin 6 (IL-6) is a critical cytokine involved in the innate immune response in sepsis and other severe infections and contributes in conjunction with other pathophysiological processes to adverse outcomes [3–6]. The modulation of IL-6 dynamics and its multiple different signalling pathways represents a potentially exciting therapeutic opportunity for severe infection given the key role for this cytokine and associated signalling [3,6,7]. The inflammatory role of IL-6 is mediated not by the classical or *trans*-presentation modes of action through a membrane bound receptor (IL6R), but through a *trans*-signalling mechanism where IL-6 binds a soluble form of the receptor that subsequently interacts with membrane bound signalling molecule, gp130, on cells [7,8].

Inhibition of both membrane and soluble IL6R using monoclonal antibodies such as tocilizumab or sarilumab (collectively known as IL-6 receptor antagonists, IL6RAs) have been successfully trialled in critically ill patients with Coronavirus Disease 2019 (COVID-19) and are now considered a standard treatment [9–11]. These drugs have the capability of attenuating all forms of IL-6 signalling, producing reductions in C-reactive protein (CRP) and other downstream inflammatory markers [11–13]. Furthermore, this beneficial effect of IL6R blockade in COVID-19 was anticipated in a causal framework analysis using genetic data [14,15]. Carriers of certain single nucleotide polymorphisms (SNPs) around and in the *IL6R* gene (the target for IL6RAs) that phenocopy IL6RA function have a reduced risk of becoming critically ill with COVID-19 [14,15].

The same *IL6R* variants have been used in Mendelian randomisation (MR) studies as instruments for alterations in IL6R function and signalling, thus providing a functional proxy of IL6RA therapy, with an extensive literature relating to the canonical SNP, rs2228145 [14,16–22]. This SNP is known to lead to reduced levels of cell surface receptor that mediates classical signalling and a coupled increase in soluble IL6R that alters the setpoint for

downstream IL-6 *trans*-signalling (**Fig 1**) [22]. A previous MR analysis suggested that reduced IL-6 signalling (modelled using rs2228145 alone) could reduce inflammation and risk of coronary heart disease [16], leading to clinical trials of monoclonal antibodies targeting either IL6R or IL-6 for coronary disease prevention [12].

We hypothesised that there may be a role for IL6R blockade in sepsis given the similarities between bacterial sepsis and critical illness in COVID-19 [23]. To test this, we undertook a two-sample MR study to assess the potential impact of IL6R blockade on sepsis, COVID-19, as well as risk of infection in the absence of sepsis.

## Methods

### Study design

In this study, we aimed to perform a two-sample MR study in order to proxy the effect of IL6R blockade on sepsis and other infections. **Fig 1** shows the overall study design, comparing our analysis with a randomised controlled trial of IL-6 receptor antagonist therapy.

### Included populations

For our main outcomes, we used UK Biobank, a large UK adult volunteer cohort described in detail elsewhere [24]. UK Biobank has linked genetic and physical data with direct links to national healthcare datasets. For secondary outcomes, we utilised FinnGen (Round 6), a large prospective cohort study in Finland, linked to electronic health record data [25]. For measurement of COVID-19 outcomes, we included data from the COVID-Host Genetics Initiative (HGI) (Round 7), a large meta-analysis of multiple studies including participants with COVID-19 [26].

Finally, for additional data on sepsis survival, we utilised summary statistics from a previous genome-wide association study (GWAS) on survival from sepsis, which included data from the GaINS and GenOSept consortium [27]. Full details of each cohort, inclusion criteria, and genetic quality control are available in **S1 Text.**

### Definition of outcomes

Our primary outcomes were the incidence of sepsis, sepsis requiring critical care utilisation, and 28-day mortality after an episode of sepsis or sepsis requiring critical care admission, measured in UK Biobank.

For secondary outcomes, we included (a) a set of 9 other common infections that present to primary or secondary healthcare (**S2 Table**) and (b) COVID-19 infection, as a comparator.

Admissions with sepsis were identified in UK Biobank in ICD-coded linked secondary care data. ICD-10 codes A02, A39, A40, and A41 were used to identify sepsis, in line with recent literature [28]. Cases were included if the code was in the primary or secondary diagnostic position in Hospital Episode Statistics (HES) data (or similar datasets in the devolved nations), provided by UK Biobank. We did not include self-reported cases, or cases only occurring in primary care. To ensure no contamination with COVID-19-related codes, we excluded codes for sepsis that occurred after 1 February 2020.

Other infections were defined similarly and included by the presence of ICD-10 codes, derived by two authors (FH and DA), with a code list available in the **S2 Table**. Codes were derived from recent publications but altered to reflect 3-digit ICD-10 coding available in HES. Controls were defined by the absence of the ICD code.

For the definition of critical illness related to infection, we utilised critical care admission data provided in HES. We considered any critical care episode during the index infection

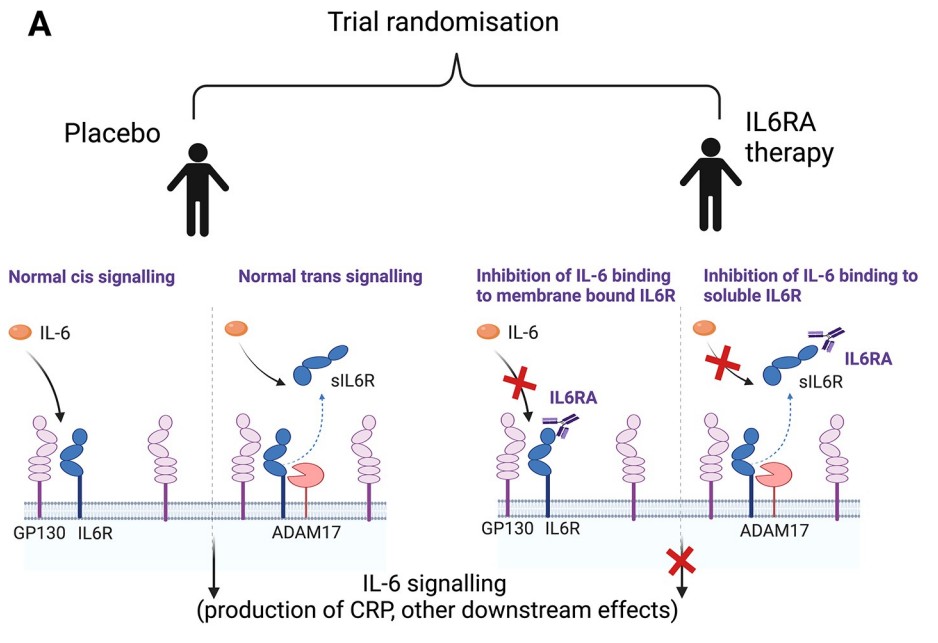

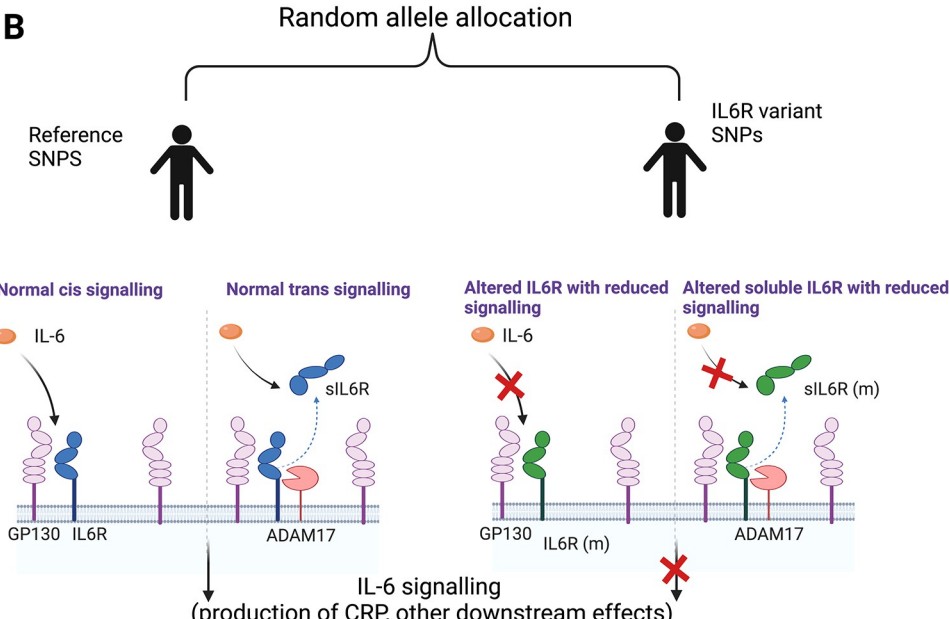

**Fig 1. Section A represents a randomised trial of IL6RA therapy with both normal IL-6 signalling and the effect of intervention.** Section B represents the use of SNPs in *IL6R* to act as a proxy for intervention. We identify variants within *IL6R* associated with reduced CRP as a marker of functional IL6R blockade to aid the interpretation of MR analysis. As these SNPs are within IL6R, we make the assumption that these SNPs have an effect through alteration of IL6R (either through modification of the protein itself or altering quantity of the protein). Image made using Biorender.com. ADAM17, A disintegrin and metalloprotease 17; CRP, C-reactive protein; GP130, Glycoprotein 130; IL-6, interleukin 6; IL6R, interleukin 6 receptor, IL6RA, interleukin 6 receptor antagonist; IL6R (m), modified interlekin 6 receptor; MR, Mendelian randomisation; sIL6R, soluble interleukin 6 receptor; SNP, single nucleotide polymorphism.

admission as a critical care admission related to that infection. Controls were defined as all other participants of UK Biobank. As HES data are only available for English participants of UK Biobank, we excluded all participants who were not recruited at a recruitment centre in England for the critical care portion of the study.

We generated 28-day survival outcomes for sepsis (with or without critical care admission) and pneumonia (only in those admitted to critical care). Dates of admission to hospital were extracted from the HES data and matched to national registry death data supplied by UK Biobank.

In order to attempt to avoid issues of bias relating to the structure of the data included in primary analysis (in particular, collider bias where selection on case status can induce associations between traits related to case status), we included all participants in our analysis of sepsis risk. For our analysis, we chose to use the whole UK Biobank population as a control for each outcome, even for the critical care related outcomes (e.g., we compared critically ill patients with sepsis versus the whole population, rather than critically ill patients with sepsis versus all patients with sepsis).

## GWAS of infection outcomes

To generate summary statistics for downstream MR, we performed a case–control GWAS on each infection outcome using regenie v2.2.4 on all UK Biobank participants of European ancestry (see critical care exclusions above) [29]. We used the in-house MRC-IEU GWAS pipeline (version 2) to quality control our data [30]. Full details of this are published elsewhere [30], with further details in **S1 Text**.

## Mendelian randomisation—Definition of instruments

Our *IL6R* instruments were selected based on a recent MR study that performed a meta-analysis of a high-sensitivity CRP (hsCRP) GWAS of 522,681 European individuals from the Cohorts for Heart and Aging Research in Genomic Epidemiology (CHARGE) Consortium and UK Biobank [31].

Conceptually, our genetic instrument is similar to the action of anti-IL6R monoclonal antibodies (e.g., tocilizumab) that lead to inhibition of IL6R signalling by blocking both IL6 classical and *trans*-signalling, although the effect of inhibition is much reduced with our genetic instrument compared to therapeutic IL-6 inhibition [16–19,32,33].

Specifically, we aimed to assess the effect of decreased activity of the IL6/IL6R pathway, modelled using independent ($r^2 < 0.1$) variants within 300 kb of *IL6R* in order to proxy IL6RA effect. We call this our *cis*IL6R instrument. We weighted this variant on the effect on hsCRP, as justified in **S1 Text**, and in line with previous analyses [17–19]. For simplicity, we henceforth refer to this exposure as "IL6R blockade."

It is recognised that this is an oversimplification of the IL6 pathway, with evidence that effects on health outcomes are mediated by classical and *trans*-signalling in differing ways and that the effect of our instrument may not act in the same way as IL6RAs [8]. In our sensitivity analyses (below), we explore other ways of defining our exposure, including alternative weighting strategies. In total, 26 variants were included. The minimum F-statistic of an included variant was 31.1 [34]. All SNPs were available in the GWAS performed above. The included SNP list is available in **S1 Table**.

## Statistical analysis

For our main analysis, we identified and extracted SNPs (or proxies, for secondary cohorts) from our outcome GWAS and performed two-sample MR using harmonised SNPs on each

outcome in term. MR estimates from each SNP were generated and then meta-analysed by inverse variance weighting (IVW) using first-order weights [21]. Analyses were performed using the *TwoSampleMR* package (version 0.5.6) in R (version 4.1.3) [35].

For the UK Biobank and FinnGen outcomes, we performed fixed effects meta-analysis across each specific infection to generate summary estimates for each condition using the R package *meta*.

### *cis*CRP variants

Our primary instrument includes only variants around *IL6R*, weighted by their effect on CRP, as a readout of IL-6 function. However, this does not imply that CRP itself is causal. In order to understand whether CRP might be the causal target, we undertook a subsequent analysis using four well-understood *cis*CRP-related variants from a recent study (**S1 Table**) [36]. These variants are highly likely to alter CRP through a pathway independent of IL-6 down-regulation, and, therefore, evidence of any MR effect would support CRP as a potential causal mechanism and also provide additional genetic support for targeting of this pathway. These variants were weighted using the same UK Biobank-CHARGE meta-analysis of hsCRP [37].

After weighting, we performed two-sample MR analyses using this instrument on each outcome in turn, comparing results to that generated using the *cis*IL6R instrument.

### *cis*gp130 instruments

gp130, also known as *IL6ST*, is the other component of the IL-6 receptor, although it is also present in other cytokine receptors. Candidate gene analyses have suggested variation in gp130 has phenotypic consequences [38], although this is less established than the association with variation at *IL6R*. To explore this further, we generated instruments for gp130 plasma protein levels using a recent GWAS of plasma protein levels from the DECODE ($n = 35,287$) consortium. [39] We extracted independent ($r^2 < 0.1$) SNPs that were associated with levels of gp130 and that were within 300 kb of gp130 for use in downstream MR analyses on the infection outcomes using the TwoSampleMR package. Again, MR was performed in turn on each outcome.

### Sensitivity analyses

We performed three broad types of sensitivity analysis. Firstly, we attempted to test one of the assumptions of MR (exclusion restriction) using alternative meta-analytic approaches (MR-Egger and weighted median approaches) [21], although these methods cannot falsify this restriction but can detect violations. Secondly, we tested whether variant weighting altered the results, by weighting variants solely on their effect estimates from CHARGE rather than from the UKB-CHARGE meta-analysis to avoid overfitting of data ("winners curse") [21], and ran the analysis weighted entirely on the beta-coefficients from the SNP-outcome association (e.g., unweighted by CRP), with these results then meta-analysed together by the inverse variance weighting of each SNP-outcome (beta coefficient) association.

Subsequently, we tested whether our choice of SNPs to include altered the results. Firstly, we used the R package *RadialMR* to identify SNP outliers and reran analyses with outliers excluded [40]. Secondly, we performed iterative leave-one-out analyses, where each SNP is left out of the model in turn and IVW estimates recalculated. Thirdly, we reran the analysis including only variants within 10 kb of *IL6R*, and, finally, we ran the canonical and well-described Asp358Ala SNP (rs2228145) as a single instrument [16].

As an additional sensitivity analysis to explore heterogeneity identified in our primary analysis (using *cis*IL6R SNPS), we performed clustering to identify SNPs that had a consistent effect on multiple traits [41]. In this analysis, we identified all outcomes for which we had effect

estimates for all 26 SNPs (31/35 outcomes, all except the GenOSept and GaINS consortium) and performed a form of mixture modelling (noise augmented directional clustering) to identify SNPs that had correlated effects on different traits and cluster them. This was performed using the *navmix* package in R using default settings (K = 10) on standardised associations between traits. Subsequently, we performed MR within each cluster to identify any evidence of heterogeneity [41].

### Reporting guidelines

This study is reported in line with the STROBE-MR guidance, with the checklist available in the Supporting information (**S1 STROBE Checklist**) [42].

### Ethics

Most of the data in this study were publicly available and nonidentifiable. Therefore, no ethical approval was required to access it. Access to UK Biobank was arranged by the UK Biobank IDAC. This study was performed under application number 56243.

## Results

### Identification of sepsis cases in UK Biobank

In UK Biobank, we identified 11,643 cases of sepsis, with 474,841 controls of European ancestry. A total of 1,896 patients died within 28 days of admission, with 484,588 controls. A total of 1,380 patients had critical care admission with sepsis, with 429,925 controls (including only UK Biobank participants in England). Approximately 347 patients died within 28 days of critical care admission, leaving 431,018 controls. We subsequently performed a case–control GWAS for each of these outcomes, with links to quantile–quantile plots and Manhattan plots available in **S2 Table,** as well as details of other included infections in **Table 1**.

### Mendelian randomisation

We performed IVW MR on each outcome in turn. As our instruments are weighted by hsCRP, odds ratios (OR) are on the scale of natural log hsCRP decrease. For reference, we call this IL6R blockade, with ORs of more than one representing increasing risk with greater interference and ORs of less than one representing decreasing risk.

For our primary outcome—sepsis—we identified a severity-dependent effect, with evidence suggesting that IL6R blockade is increasingly protective with more severe disease. The OR for sepsis was 0.80 (95% confidence interval (CI) 0.66 to 0.96); for 28-day death in sepsis was 0.74 (95% CI 0.47 to 1.15), for sepsis requiring critical care admission was 0.48 (95% CI 0.30 to 0.78), and for 28-day death in sepsis requiring critical care admission was 0.37 (95% CI 0.14 to 0.98) (Fig 2A and Table 1).

These severity-dependent effects were mirrored when we performed IVW MR on COVID-19-related outcomes from the COVID-HGI, with evidence suggesting that IL6R blockade is more protective from critical respiratory illness, defined as those who required respiratory support or who died during hospitalisation (OR 0.69, 95% CI 0.57 to 0.84) than from hospitalisation alone (OR 0.83, 95% CI 0.74 to 0.93). Notwithstanding uncertainty around the point estimates, MR estimates for IL6R blockade were larger in sepsis than in COVID-19, for the comparable disease severity.

We performed IVW MR to investigate the association of IL6R blockade on the odds of infection in the absence of sepsis within UK Biobank. In line with trial and registry literature, there was evidence suggesting an increase in susceptibility to evaluated infections. The largest

**Table 1.  IVW meta-analysis of MR estimates of IL6R blockade for UK Biobank and COVID-19 HGI outcomes.**

|  | OR (95% CI) | *P* value | Cases / controls |
|---|---|---|---|
| Sepsis |  |  |  |
| Sepsis (all admissions) | 0.8 (0.66–0.96) | 0.019 | 11,643 / 474,841 |
| Sepsis (28-day death) | 0.74 (0.47–1.15) | 0.180 | 1,896 / 484,588 |
| Sepsis (critical care) | 0.48 (0.3–0.78) | 0.003 | 1,380 / 429,985 |
| Sepsis (28-day death in critical care) | 0.37 (0.14–0.98) | 0.046 | 347 / 431,018 |
| COVID-19 |  |  |  |
| COVID-19 (All cases vs. population) | 0.93 (0.89–0.98) | 0.009 | 159,840 / 2,782,977 |
| COVID-19 (Hospitalised) | 0.83 (0.74–0.93) | 0.002 | 44,986 / 2,356,386 |
| COVID-19 (Severe respiratory vs. population) | 0.69 (0.57–0.84) | <0.001 | 18,152 / 1,145,546 |
| Infection |  |  |  |
| URTI | 1.67 (1.13–2.48) | 0.010 | 2,795 / 483,689 |
| Endocarditis | 1.66 (0.88–3.15) | 0.120 | 1,080 / 485,404 |
| Cholecystitis | 1.53 (1.12–2.08) | 0.007 | 4,052 / 482,432 |
| UTI | 1.32 (1.17–1.5) | <0.001 | 21,958 / 464,256 |
| Osteomyelitis | 1.27 (0.88–1.82) | 0.207 | 4,836 / 481,648 |
| Appendicitis | 1.21 (0.93–1.58) | 0.153 | 4,604 / 481,880 |
| Cellulitis | 1.2 (1.02–1.41) | 0.028 | 12,196 / 474,288 |
| LRTI | 1 (0.83–1.22) | 0.960 | 14,135 / 472,349 |
| Pneumonia | 0.97 (0.86–1.1) | 0.642 | 22,567 / 463,917 |
| Sepsis | 0.8 (0.66–0.96) | 0.019 | 11,643 / 474,841 |
| Other critical care outcomes |  |  |  |
| Pneumonia (critical care) | 0.69 (0.49–0.97) | 0.034 | 2,758 / 428,607 |
| LRTI (critical care) | 0.51 (0.23–1.12) | 0.095 | 585 / 430,780 |
| Pneumonia (28-day death in critical care) | 0.5 (0.23–1.08) | 0.077 | 803 / 430,820 |

CI, confidence interval; COVID-19, Coronavirus Disease 2019; HGI, Host Genetics Initiative; IVW, inverse variance weighted; LRTI, lower respiratory tract infection; MR, Mendelian randomisation; OR, odds ratio; URTI, upper respiratory tract infection; UTI, urinary tract infection.

effect sizes observed were for upper respiratory tract infection (URTI) (OR 1.67; 95% CI 1.13 to 2.48) and for endocarditis (OR 1.66; 95% CI 0.88 to 3.15). For respiratory infections—lower respiratory tract infection (LRTI) and pneumonia—we saw no strong evidence of effect (OR approximately 1 for both) (Fig 2C and Table 1).

For both of these respiratory infections, despite the largely null effect with incidence of disease, IL6R blockade was associated with decreased odds of critical care admission (Fig 3A and Table 1), with effect estimates concordant with the estimates from sepsis requiring critical care admission: OR for pneumonia requiring critical care admission was 0.69 (95% CI 0.49 to 0.97) and the OR for LRTI requiring critical care admission was 0.51 (95% CI 0.23 to 1.21).

## Follow-up analysis

We were able to extract parallel GWAS results from Round 6 of the FinnGen consortium (S3 Table) and to rerun all analyses. The frequency of incident cases differed across infection between studies. This was most notable for LRTI (frequency of incident cases 2.9% UK Biobank compared to 1.05% FinnGen). We identified three sepsis outcomes to compare: one combined one (all cases of ICD-coded sepsis) and two specific ones (subsets of the combined outcome). For the combined sepsis outcome, the frequency of incident cases was 54% higher in FinnGen (2.4% UK Biobank, 3.7% FinnGen), while the frequency of mortality at 5 years—

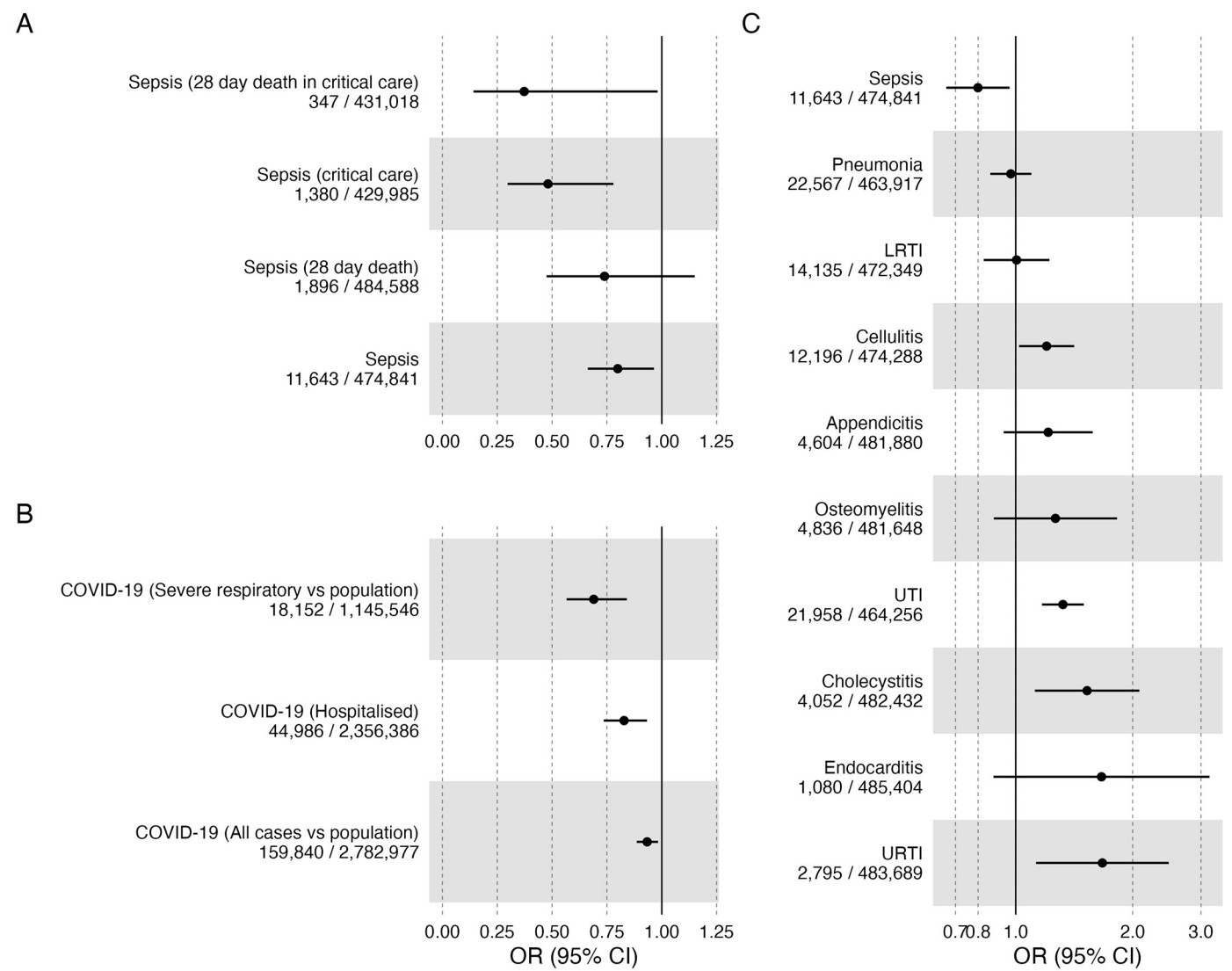

**Fig 2.** IVW MR estimates (ORs) with 95% CIs for IL6R blockade and each outcome (A: Sepsis, B: COVID, C: other infections) in UK Biobank. CI, confidence interval; COVID-19, Coronavirus Disease 2019; IVW, inverse variance weighted; LRTI, lower respiratory tract infection; MR, Mendelian randomisation; OR, odds ratio; URTI, upper respiratory tract infection; UTI, urinary tract infection.

the only mortality figure available within FinnGen—was 20.2% as opposed to 37.6% within UK Biobank.

For the specific sepsis definitions available in FinnGen—sepsis due to pneumonia and streptococcal sepsis—mortality data were not available, but these were rarer than the combined sepsis outcome (0.5% and 0.2%, respectively).

For the combined FinnGen sepsis outcome, the OR was close to the null: OR 0.98 (95% CI 0.79 to 1.21). However, when focusing on the two specific outcomes of streptococcal sepsis and sepsis due to pneumonia, effect estimates were similar to those using UK Biobank: OR 0.79 (95% CI 0.48 to 1.31) and OR 0.63 (95% CI 0.29 to 1.35), respectively, although with more uncertainty due to a smaller sample size (**Fig 3C**).

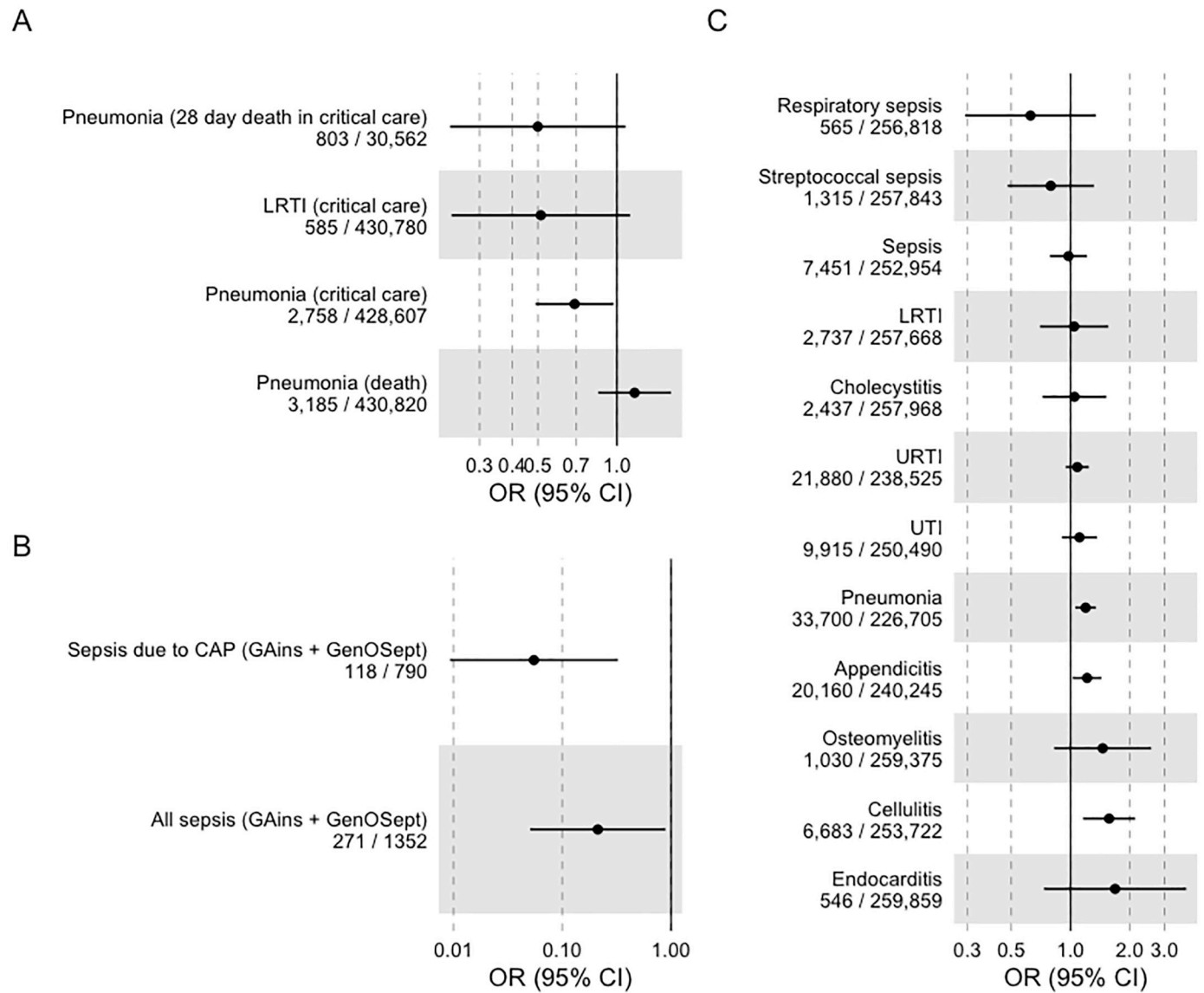

**Fig 3.** IVW MR estimates of IL6R blockade with 95% CIs for (A) respiratory infection, (B) survival from sepsis related to critical care admission, and (C) FinnGen replication cohort. CI, confidence interval; IVW, inverse variance weighted; LRTI, lower respiratory tract infection; MR, Mendelian randomisation; OR, odds ratio; URTI, upper respiratory tract infection; UTI, urinary tract infection.

Across UK Biobank and FinnGen, effect estimates were largely consistent in direction for all infections except pneumonia, although effect estimates were generally smaller. The meta-analysed summary estimate for sepsis, utilising UK Biobank and the main FinnGen sepsis outcome, had an OR of 0.86 (0.74 to 0.99), when meta-analysing with streptococcal sepsis in FinnGen, the OR was 0.80 (0.67 to 0.95), and when meta-analysing with respiratory sepsis in FinnGen, the OR was 0.79 (0.66 to 0.95). FinnGen-specific and meta-analysed summary effects are available in S4 Table, with a summary forest plot in **S1 Fig**.

### Additional survival outcomes

Additional survival outcomes were available from two previous GWAS performed in the GAiNS and GenOSept consortium, both of which recruited patients with sepsis in critical care but also included a subset of patients with confirmed CAP. In both studies, case–control GWAS was performed with the outcome of 28-day survival.

In both studies, we performed IVW MR and effects were concordant with our primary data (**Fig 3B),** with a summary OR of 0.22 (95% CI 0.04 to 1.31) for sepsis and 0.06 (95% CI 0.01 to 0.55) for the CAP subset. These estimates were imprecise, given the number of included cases and controls.

### Alternative genetic instruments

Although our genetic instrument includes variants around *IL6R* (acting as a proxy for the target drug effect), we weighted it by the effect on hsCRP as this represents an appropriate read-out of IL6R function and the effect on downstream IL-6 signalling. However, this does not mean that hsCRP is itself necessarily part of the causal pathway, and it may simply represent a measurable marker of IL6R blockade. It is also plausible that the beneficial effect of IL-6 blockade in sepsis is actually mediated by reductions in hsCRP. In an attempt to investigate this potential mediation and elucidate a potential mechanism of effect of IL6R blockade, we reran analyses using established *cis* variants around *CRP* known to be associated with circulating hsCRP levels.

In IVW MR analyses using the *cis*CRP instrument, we found evidence that CRP may be a part of the IL6R blockade effect, with evidence of reduced odds of sepsis and sepsis-related mortality: OR for sepsis outcome (0.91, 95% CI 0.82 to 1), OR for sepsis critical care admission (0.88, 95% CI 0.65 to 1.17), OR for death (0.72, 95% CI 0.59 to 0.93), OR for critical care death (0.58, 95% CI 0.32 to 1.04) (**Fig 4** and **S6 Table**). This effect was also seen in FinnGen: OR for combined sepsis outcome (0.76, 95% CI 0.59 to 0.98), with similar results for other sepsis definitions. These effects were generally smaller in size than those seen using the *cis*IL6R instrument, with less evidence of a graduated effect with severity as seen with the *cis*IL6R instrument. Effect estimates were again broadly similar (although imprecise) for protection against critical infection (e.g., OR for critical care admission with LRTI 0.77, 95% CI 0.49 to 1.21).

Outside of sepsis, there was little evidence of correlation between IVW MR results from *cis*CRP- and *cis*IL6R-related instruments (Pearson's R 0.06), and, in addition, there was no strong evidence for an association with COVID-19 outcomes.

Subsequently, we investigated *cis* variants in gp130 (also known as IL6ST). Using a recently performed large-scale GWAS of gp130 plasma protein level (DECODE, $n = 35,287$), we identified 24 independent ($r^2 < 0.1$) SNPs (S1 Table). IVW MR was performed as described above in this analysis (reported in **S7 Table** and **S2 Fig**).

The OR for sepsis with each SD increase in gp130 levels was 0.97 (95% CI 0.91 to 1.04), with similar estimates for the sepsis outcome in FinnGen (OR 1.04, 95% CI 0.96 to 1.14). Results for other infection outcomes were similar, with little clear evidence of an effect. The strongest evidence of any effect was in death from critical care pneumonia (OR 1.58; 95% CI 1.14 to 2.19) and death from critical care sepsis (OR 1.57; 95% CI 1.06 to 2.34), although effect estimates on death from hospitalised sepsis were close to the null (OR 0.95; 95% CI 0.81 to 1.13).

### Noise-augmented clustering

As there was visual evidence of potential clustering of SNP effect estimates for our primary instrument (26 *cis IL6R* SNPs), we performed a form of mixture modelling (noise-augmented directional clustering) to identify clusters of instruments.

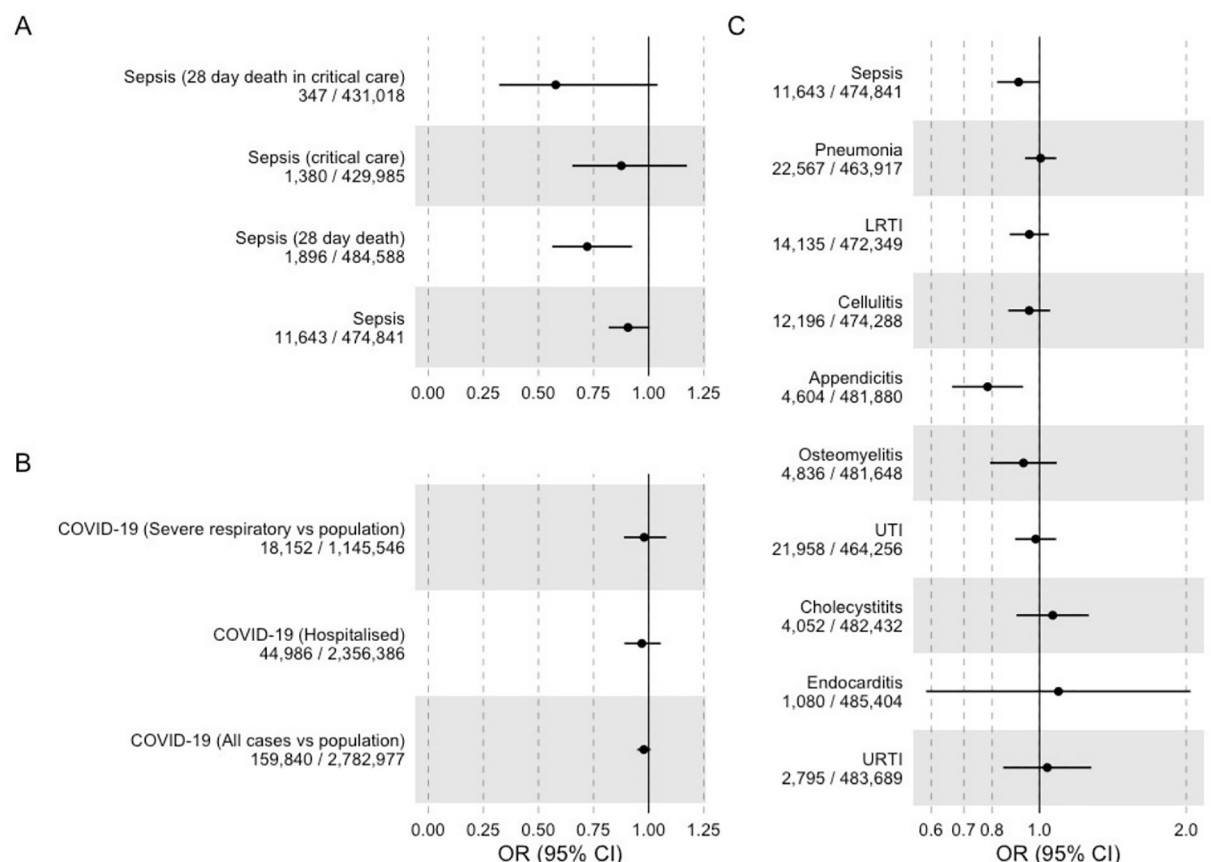

**Fig 4.** IVW MR effect estimates for CRP from *cis*CRP variants and each outcome (A: Sepsis, B: COVID, C: other infections). CI, confidence interval; COVID-19, Coronavirus Disease 2019; CRP, C-reactive protein; IVW, inverse variance weighted; LRTI, lower respiratory tract infection; MR, Mendelian randomisation; OR, odds ratio; URTI, upper respiratory tract infection; UTI, urinary tract infection.

In this analysis, we identified one cluster of 14 SNPs (heatmap shown in **S3 Fig**) with all other SNPs excluded as noise (included SNPs listed in **S8 Table**). Subsequently, we performed MR using just these instruments (**S9 Table** and **S4 Fig**). These analyses showed similar results to our primary analysis, although effect estimates were generally larger, and *p*-values generally lower.

## Sensitivity analyses

We performed a range of sensitivity analyses that did not alter the interpretation of primary results. Firstly, where possible, we tested some of the assumptions of MR using different meta-analytic techniques [21]. For these analyses, we performed meta-analysis using MR-Egger and weighted median approaches, which test the exclusion restriction assumption. Weighted median analyses were broadly similar to IVW results, but MR-Egger estimates were more imprecise, with nearly all confidence intervals for outcomes in UK Biobank, FinnGen, and COVID-HGI crossing the null. These results are shown in **S5 Fig** and **S10 Table**, while representative scatter plots for two representative outcomes (sepsis and critical care admission with sepsis) are available in **S6 Fig**.

In a second set of sensitivity analyses, we tested whether the way we designed and weighted our instrument materially affected the results. Firstly, we weighted our exposure SNPs using

betas solely from the CHARGE consortium [34] (rather than using a combined CHARGE-UK Biobank meta-analysis; **S11 Table**). These results were similar (although more imprecise, due to the smaller sample size of the original GWAS). Secondly, we ran completely unweighted analyses (**S12 Table** **and S7 Fig**), which simply meta-analysed the effect of each SNP-outcome association, using IVW of the beta-coefficients at each SNP. These results were again concordant with our primary analysis, although were highly imprecise (given the small individual effect of each SNP).

In a third set of sensitivity analyses, we tested whether our selection of *IL6R* SNPs influenced results. Firstly, we identified and removed outliers using Radial MR and reran analyses (**S8 Fig** **and S13 Table**). We identified no alteration in our primary results. Alongside that, we performed leave-one-out analyses, removing individual SNPs from the analyses and rerunning IVW MR. For ease, we show this for the first four outcomes only in **S9 Fig**, which show the removal of any individual SNP does not alter the main results. Finally, we restricted analyses to the seven SNPs within 10 kb of *IL6R* (**S14 Table**) and the rs2228145 SNP, which has a known functional effect on IL6R and downstream effects [36] (**S10 Fig** **and S15 Table**). Again, these effects were all concordant with our primary results, although with reduced power as expected by removing multiple SNPs from the meta-analysis.

## Discussion

In this study, we provide evidence from multiple, independent data sources that blockade of IL-6 signalling pathways is likely to be protective against the development of sepsis. There is evidence across both UK Biobank and FinnGen that the apparent protective effect of functional IL6R blockade increases with increasing severity of illness with potential protective effects on short-term death in sepsis and critical care admission with sepsis. This effect was similar (although slightly larger) than estimates relating to severe COVID-19, where IL6RA have already been shown to improve mortality [11]. In contrast, functional IL6R blockade appears to show evidence suggesting increases in susceptibility to infections, matching trial and registry data. We also found evidence that IL6R blockade may be protective in critical illness in respiratory infection, where effect estimates were similar to those seen in sepsis and consistent with the COVID-19 data. Given the similarities between COVID-19 pneumonitis, sepsis, and bacterial respiratory infection and with commonalities in underlying pathophysiology, this suggests IL6RA as a potentially broad therapeutic target for patients unwell with critical infection. However, estimates for some phenotypes (e.g., death from critical care sepsis) were necessarily imprecise given the small sample size, although estimates were all concordant in direction.

Our analysis using *cis*CRP instruments had similar findings, although effect estimates were generally smaller. This suggests that one potential route by which IL6R blockade reduces the odds of severe sepsis is by reducing CRP, although this remains a hypothesis. Given the ongoing development of therapeutics that target *trans*-specific IL-6 signalling and therapeutics that target CRP itself, this may represent a future avenue in sepsis therapeutics [43,44]; however, this does not alter the interpretation of the primary IL6R blockade-related finding here. Additionally, the concordance of MR effect estimates between *cis*CRP and *cis*IL6R genetic variants provides confidence in our primary analysis and strongly supports the role of IL-6 signalling in sepsis. We did not identify a robust effect on sepsis when instrumenting gp130, although there is much less evidence that genetic variation at this locus affects IL-6 signalling [45].

To our knowledge, previous literature has not identified an association between variants proxying IL6R blockade and sepsis, although multiple studies have identified associations (in line with our estimates), suggesting these *IL6R* variants increases the risk of infection, which match randomised trial data [31,46]. One trial was considering the use of IL6RA in paediatric

sepsis (NCT04850443), but this was halted due to lack of funding. Furthermore, randomised trials have been performed aiming to remove IL-6 and other cytokines by using extracorporeal haemoadsorption devices [47]. In the largest trial (97 evaluable patients), the use of haemoadsorption in patients with severe sepsis was not associated with any reduction in plasma IL-6 levels and had no effect on mortality once adjusting for comorbidities (hazard ratio 1.67, 95% CI 0.77 to 3.61) [48]. Given that this device is untargeted, did not successfully reduce IL-6 levels and the clinically relevant complications associated with usage of extracorporeal haemoadsorption, it is hard to interpret this evidence in evaluation of targeted IL-6 down-regulation by either genetic variation or IL6RA.

Despite a large sample size and multiple independent sources of data, this study has weaknesses. Firstly, we rely on diagnostic coding for infections for both FinnGen and UKB, although we utilise clinically diagnosed cases in GenOSept and GAiNS. As sepsis is heterogenous and poorly characterised, these likely reflect slightly different pathological processes. Supporting this, some estimates in our replication cohort varied with the definition of sepsis, with much larger effects in those with specific sepsis codes (e.g., streptococcal sepsis) than with the main code. However, this smaller effect may be due to the FinnGen sepsis cases being of lower severity than the UK Biobank cohort, reflected in the higher frequency of incident cases of sepsis, and reduced severity of disease. Given our results show IL-6 activity has both protective (from infection) and detrimental (development of sepsis) effects, even small changes to include less sick populations will likely greatly reduce the effect size, as seen in the FinnGen data and demonstrated in studies and simulations of phenotypic misclassification. Importantly, despite these differing study designs and approaches, nearly all estimates converged towards a protective effect of IL-6 inhibition.

These results are subject to other factors that potentially complicate the translation of applied epidemiological analysis into clinical trials. This is particularly acute with respect to the potential definition of a population that might benefit from IL-6 inhibition in the context of the severity of sepsis, but also the possible risks according to other infection risk (especially in the frail). This remains a question not addressed directly by the work here, but of potentially great importance if IL6RA are to be considered as interventions for acute outcomes.

Related to this, a major challenge also exists in interpreting potential biases induced with the analysis of sepsis, which is a product of case status—i.e., of severe infection. As the down-regulation of functional IL6R potentially leads to increased risk of infection, analyses of genetic variants related to progression to severe infections (e.g., sepsis) has the potential to be biased. In extreme cases, collider bias could lead to unpredictable biases on effect estimates, if the genetic effects on incidence of sepsis were very large. For that reason, we undertook GWAS using the entire cohorts, rather than performing a "case-only" analysis. This potentially avoids the impact of this specific problem but leaves interpretation subject to results based on control status including mild infection. As a consequence, this may mean that the size effect estimates here (e.g., for apparent reduced odds of severe sepsis) should not be interpreted as the same as those potentially occurring with the use of IL6RA therapy in real life. Despite this, it is reassuring that our effect estimates are similar in size and direction to COVID-19 effect estimates, where a large and clinically important effect was identified (3% to 4% absolute reduction in mortality for those hospitalised with COVID-19) [11]. In the absence of a trial, these types of interpretation complication remain difficult to escape.

Focusing on variants within or near the *IL6R*, it is difficult to completely rule out pleiotropic effects (e.g., these SNPs acting to reduce risk of sepsis by another, unrelated mechanism). IL-6 is highly pleiotropic, and variants at *IL6R* have been shown to have effects on a wide range of clinical and biochemical traits [49], with a reduction in the odds of a number of cardiometabolic traits with genetically proxied IL-6 down-regulation [50]. It is therefore difficult to be confident of the exact mechanism of protection that we identify. However, given the

concordance in protection from severe COVID-19 identified in our study, randomised trial data matching that using genetic proxies for IL6R blockade, and the location of these SNPs, we can have some confidence that the effect is driven by alterations in IL6R.

Finally, in common to all genetic studies, translation of (small) effects due to germline variation into the context of a severely ill population receiving a large, time-limited intervention requires detailed thought. Not only does a focus on variants within or near the *IL6R* fail to guarantee the absence of complications generated as a result of pleiotropy; most of our estimates relate to protection from the odds of sepsis events. Any potential trial is likely to enrol patients who already have sepsis, and given the difficulties in predicting sepsis and the practicalities of administering IL6RA, differences between those circumstances and results here may appear. Furthermore, in clinical trials of COVID-19 outcomes, participants also received corticosteroids with some evidence of an interaction effect between corticosteroids and IL6RA [11]. Although the risk of nosocomial infection was not high in clinical trials in COVID-19 (<1% in both REMAP-CAP and RECOVERY; [9,10]), careful evaluation of the potential harms of IL6RA in a population with infection will be required in trial design, given the double-edged nature of IL6 inhibition.

## Conclusions

Although we should be appropriately sceptical of any novel therapeutics in sepsis, given the failure to identify any successful agents despite 30 years of research, the unique conditions surrounding this work suggest that IL6R blockade may be a useful approach. Firstly, the use of MR to identify potential therapeutic targets is supported by a large amount of empirical evidence, replicating both positive and negative trial effects [20,51]. Secondly, our specific technique of using *IL6R* variants as an exposure has been utilised before, with randomised trial data matching MR estimates [12,16]. Finally, the similarities between COVID-19 and sepsis pathophysiology are clear, with robust trial data supporting the role of IL6RA in COVID-19 [11]. Against expectations, adjusted rates of secondary infection in the IL6RA-treated population were similar (OR: 0.99, 95% CI: 0.85, 1.16) than those in comparator arms in the recent WHO meta-analysis, which provides additional comfort that IL6RA are safe to use in the critically ill [11].

Our data are therefore suggestive that functional down-regulation of IL6 may have beneficial effects in improving sepsis outcomes. Given the previous data linking genetic variation in *IL6R* with trial outcomes of IL6RA, and the biological plausibility of effect, these data support trialling IL6RA in sepsis.

## Supporting information

**S1 STROBE Checklist. STROBE Checklist for this study.**
(DOCX)

**S1 Fig. Odds ratios for the effect of IL6R blockade for each outcome in FinnGen (red), UK Biobank (blue), and meta-analysed across both (black).** Results generated by IVW MR.
(PNG)

**S2 Fig.** Effect estimates (odds ratios) generated by inverse variance weighted Mendelian randomisation for plasma protein levels of gp130 for (A) sepsis-related outcomes, (B) COVID-19-related outcomes, and (C) other UK Biobank–related outcomes.
(PNG)

**S3 Fig. Heatmap of each variants effects on multiple traits identified by noise-adjusted clustering.** This includes 14 SNPs, with only one cluster identified.
(PNG)

**S4 Fig.** Effect estimates (odds ratios) generated by inverse variance weighted Mendelian randomisation for the 14 *cis*IL6R SNPs identified as a cluster by noise-augmented clustering for (A) sepsis-related outcomes, (B) COVID-19-related outcomes, and (C) other UK Biobank–related outcomes.
(PNG)

**S5 Fig.** Effect estimates (odds ratios) generated by weighted median and MR-Egger meta-analytic approaches for (A) sepsis-related outcomes, (B) COVID-19-related outcomes, and (C) other UK Biobank–related outcomes.
(PNG)

**S6 Fig. Scatter plots for each SNP-outcome and SNP-exposure association for two outcomes: sepsis and critical care admission with sepsis.** Each line represents the summarised effect estimate from a meta-analytic approach.
(PNG)

**S7 Fig.** Odds ratios for sepsis generated by meta-analysing the SNP-outcome association (i.e., unweighted analysis) for 26 *cis*IL6R variants using inverse variance weighting and the effect on (A) sepsis-related outcomes, (B) COVID-19-related outcomes, and (C) other UK Biobank infection outcomes.
(PNG)

**S8 Fig. Odds ratios for the effect of IL6R blockade for each outcome with outlying SNPs identified by Radial MR removed in FinnGen (red), UK Biobank (blue), and meta-analysed across both (black).** Results generated by IVW MR. Red represents the MR estimate with outliers removed, black the primary MR estimates.
(PNG)

**S9 Fig.** Inverse-variance weighted MR effect estimates (betas) when leaving out one SNP iteratively (leave-one-out analyses) for (A) critically unwell sepsis cases, (B) level 3 sepsis cases, (C) sepsis cases, and (D) sepsis-related mortality.
(PNG)

**S10 Fig.** Odds ratios generated by the Wald ratio for the Asp358Ala (rs2228145) SNP association with (A) sepsis-related outcomes, (B) COVID-19-related outcomes, and (C) all other UK Biobank outcomes.
(PNG)

**S1 Text. Supplementary methods.**
(DOCX)

**S1 Table. Included SNPs for the functional IL-6 down-regulation exposure (*cis*IL6R), and the *cis*CRP exposure.** Effect alleles here are for increasing CRP, but in the manuscript, we reverse these to show the effect of down-regulation.
(XLSX)

**S2 Table. Details of included UKB GWAS details on methodology of GWAS are in the supplementary methods.**
(XLSX)

**S3 Table. Other GWAS included in this study.**
(XLSX)

**S4 Table. Meta-analytic results combining UKB and FinnGen IVW MR estimates in a fixed effects model.**
(XLSX)

**S5 Table. Sepsis survival GWAS (GAiNS and GenOSept).** Odds ratios are for chance of death per lnCRP decrease.
(XLSX)

**S6 Table. Odds ratios for IVW MR estimates for each outcome with *cis*CRP exposures and are on the scale of lnCRP decrease.**
(XLSX)

**S7 Table. Inverse variance weighted MR results for gp130 plasma protein levels on each outcome.** Results are reported on the scale of one SD increase in gp130.
(XLSX)

**S8 Table. Fourteen SNPs identified as a cluster using noise-augmented clustering approach for our *cis*IL6R instrument.**
(XLSX)

**S9 Table. MR estimates generated using the 14 SNPs identified as a cluster by noise-adjusted clustering of our *cis*IL6R instrument.** Results generated by inverse variance weighting.
(XLSX)

**S10 Table. Estimates generated by alternative MR analytic methods (weighted median and MR-Egger).**
(XLSX)

**S11 Table. IVW MR estimates weighted by CRP weights from the CHARGE consortium alone.**
(XLSX)

**S12 Table. Inverse variance weighted estimates for the risk of each outcome using an unweighted analysis.** In this analysis, the raw association between each SNP and outcome is calculated and then meta-analysed, without weighting on any the SNP-exposure association.
(XLSX)

**S13 Table. Inverse variance weighted MR estimates for our main analytic approach once outliers (identified using Radial MR) excluded.** Note: Only outcomes with 1 or more outlier are included, outcomes where no outliers were identified are not shown.
(XLSX)

**S14 Table. IVW MR estimates limiting to SNPs within 10 kb of IL6R.**
(XLSX)

**S15 Table. MR Wald ratio estimates between the canonical rs2228145 / Asp358Ala SNP.**
(XLSX)

## Author Contributions

**Conceptualization:** Fergus W. Hamilton, David Arnold, Ed Moran, Alasdair MacGowan, Peter Ghazal, Nicholas J. Timpson.

**Data curation:** Fergus W. Hamilton.

**Formal analysis:** Fergus W. Hamilton, Tom Palmer, George Davey Smith.

**Funding acquisition:** Fergus W. Hamilton, Golam M. Khandaker, Peter Ghazal, Nicholas J. Timpson.

**Investigation:** George Davey Smith.

**Methodology:** Charlotte Summers, George Davey Smith, Nicholas J. Timpson.

**Project administration:** Fergus W. Hamilton, Nicholas J. Timpson.

**Resources:** Alexander J. Mentzer, Kenneth Baillie, Peter Ghazal.

**Supervision:** Nick Maskell, Alasdair MacGowan, Ruth Mitchell, George Davey Smith, Peter Ghazal, Nicholas J. Timpson.

**Validation:** Alasdair MacGowan, Nicholas J. Timpson.

**Writing – original draft:** Fergus W. Hamilton.

**Writing – review & editing:** Matt Thomas, David Arnold, Tom Palmer, Ed Moran, Alexander J. Mentzer, Nick Maskell, Kenneth Baillie, Charlotte Summers, Aroon Hingorani, Alasdair MacGowan, Golam M. Khandaker, Ruth Mitchell, George Davey Smith, Peter Ghazal, Nicholas J. Timpson.

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
