## [Editor Report · Decision Letter 0]

17 Jul 2022

Dear Dr Hamilton, 

Thank you for submitting your manuscript entitled "Therapeutic potential of IL6R blockade for the treatment of sepsis and sepsis-related death: Findings from a Mendelian randomisation study" for consideration by PLOS Medicine.

Your manuscript has now been evaluated by the PLOS Medicine editorial staff and I am writing to let you know that we would like to send your submission out for external peer review.

Please re-submit your manuscript within two working days, i.e. by Jul 19 2022 11:59PM.

Kind regards,

Beryne Odeny

PLOS Medicine

---

## [Decision Letter · Decision Letter 1]

31 Oct 2022

Dear Dr. Hamilton,

Thank you very much for submitting your manuscript "Therapeutic potential of IL6R blockade for the treatment of sepsis and sepsis-related death: Findings from a Mendelian randomisation study" (PMEDICINE-D-22-02409R1) for consideration at PLOS Medicine. 

Your paper was evaluated by an associate editor and discussed among all the editors here. It was also discussed with an academic editor with relevant expertise, and sent to independent reviewers, including a statistical reviewer. The reviews are appended at the bottom of this email and any accompanying reviewer attachments can be seen via the link below:

[LINK]

In light of these reviews, I am afraid that we will not be able to accept the manuscript for publication in the journal in its current form, but we would like to consider a revised version that addresses the reviewers' and editors' comments. Obviously we cannot make any decision about publication until we have seen the revised manuscript and your response, and we plan to seek re-review by one or more of the reviewers. 

We hope to receive your revised manuscript by Nov 21 2022 11:59PM. Please email us (plosmedicine@plos.org) if you have any questions or concerns.

We look forward to receiving your revised manuscript. 

Sincerely,

Callam Davidson,

Associate Editor 

PLOS Medicine

plosmedicine.org

Comments from the Academic Editor:

This work is based on an MR analysis of high methodological standard which uses summary data from several large datasets. Although I agree that the findings are not completely new, they do extend current knowledge on the therapeutic potential of targeting IL6R in critically ill patients, from an effect on COVID-19 to an effect on bacterial sepsis. I've noticed that the authors sometimes over-interpret their findings – for example, in the Abstract they report a protective effect of IL6R blockade on “sepsis survival in critical care (OR=0.22; 95% CI 0.04-1.31)” ignoring the fact that the 95%CI is compatible with an opposite effect of increased risk of about 30%.

Please update your title to ‘Therapeutic potential of IL6R blockade for the treatment of sepsis and sepsis-related death: a Mendelian randomisation study’.

Please add ‘GDS is a member of PLOS Medicine’s editorial board’ to your Competing Interests. 

Please structure your Abstract using the PLOS Medicine headings (Background, Methods and Findings, Conclusions).

Please ensure that all numbers presented in the abstract are present and identical to numbers presented in the main manuscript text.

Please define the primary outcomes in your Abstract Methods and Findings.

In the last sentence of the Abstract Methods and Findings section, please describe the main limitation(s) of the study's methodology.

Please include continuous line numbering throughout your manuscript. 

Please update your citations throughout to be normal script and placed within square brackets preceding punctuation. 

Thank you for including a STROBE-MR checklist. Please update the checklist to use section headings/paragraph numbers rather than page numbers (these are likely to change in the event of publication).

Please cite your STROBE-MR checklist as ‘S1 checklist’ or similar, to ensure correct hyperlinking in the event of publication.

Please confirm that the licensing terms associated with Biorender.com permit publication under CC-BY creative commons attribution (used by PLOS Medicine).

Please cite your Supplementary Methods as ‘S1 Text’ or similar (as described here: https://journals.plos.org/plosmedicine/s/supporting-information).

Please relocation the information under the ‘Data availability’ subheading in the Methods is also captured in the Data Availability Statement section of the submission form (in the event of publication, this information will be published as metadata).

Table 1 (and throughout): Please do not report P=0.000, report as P<0.001.

Please temper statements such as ‘Previous literature has not identified’ in the Discussion by adding ‘to our knowledge’.

The ‘Funding’ and ‘Conflicts of interest’ sections can be removed from the main text as all relevant information should be captured in the responses to the submission form (which will be published as metadata in the event of acceptance).

The Ethics section should be relocated to the Methods. 

The inclusion of [Internet] in your References is only required where a journal is online only (otherwise it should be removed and cited as if print, even if access was made online).

When citing preprints, please include [preprint] preceding the date of citation. If the preprint is now published, please update the reference accordingly. 

Comments from the reviewers:

Reviewer #1: Hamilton et al present a summary-level drug-target Mendelian randomisation study to investigate the effect of genetically-proxied IL6R blockade on sepsis. They present a wide range of sensitivity analysis, using different data resources and varying options in the MR-pipeline. My comments focus mainly on the Mendelian randomisation analysis.

Major

1. Looking at the scatter-plots in Supplementary Figure S3 Figure, there seem to be several clusters of instruments, one of which is even risk increasing. Is there the possibility that the cis-IL6R SNPs act via different pathways? Have you tried to cluster the IVs (e.g. using Noise-augmented directional clustering doi.org/10.1371/journal.pgen.1009975) to see if there is any evidence that the cis-IL6R SNPs form different pathways? 

2. It would be interesting to perform a look-up in MR-Base if the genetic variants making up the cisILR6 instrument are associated with any other phenotypes that may be on the causal pathway from genetic variant to outcome. Especially in case if there are different clusters (See Comment 1), it would be great to understand what may cause the heterogeneity seen in the scatter-plot.

3. The description of sensitivity analysis c) (Methods: "ran the analysis weighted entirely on the SNP-outcome association (e.g. unweighted by CRP)" Results: "Secondly, we ran completely unweighted analyses (Supplementary Table 9, Supplementary Figure 4), which simply meta-analysed the effect of each SNP-outcome association, unweighted by any downstream effect.") is unclear. Does this refer to weights in the inverse-variance weighting or to the beta-coefficients? Please clarify.

4. Please add Legends to all Figures in the Supplement.

Minor:

- Please harmonize your notation of technical terms, e.g the manuscript contains both "Two Sample Mendelian randomisation study" and "two sample Mendelian randomisation study".

- Please clarify which weighting scheme was used for IVW, first or second order weights.

- When describing the sensitivity analysis in Methods Section "Sensitivity analyses:", please stick with either text or a list containing of a) b) and c). What list does the "c) ran the analysis weighted entirely" refer to?

- Please introduce abbreviations like IVW and MR at first use and then stick with abbreviation throughout the text. Similarly, in the Figure and Table Legends please be consistent if you use abbreviations or not.

Reviewer #2: This manuscript reports the results of a study using pooled genetic data from five databases undertaken to evaluate the potential role of IL-6 receptor blockade on the clinical outcomes of patients with sepsis. The authors have used Mendelian randomization - a technique that leverages genetic variability under the assumption that such variability occurs randomly in a cohort of patients who subsequently develop a medical condition. They use CRP levels as a surrogate for IL-6 agonist activity, and show that SNPs associated with reduced CRP expression correlate with a reduced risk of death. The effect is most striking in patients with severe disease, and replicated across multiple sensitivity analyses. They conclude that IL-6 receptor blockade may be beneficial in sepsis, and recommend an RCT to test this hypothesis.

Mendelian randomization is a recent, and potentially powerful tool to explore causal inference in complex diseases, and the merit of this paper is less in the specific example of IL-6 receptor antagonism, for which compelling data exist in the COVID-19 literature, and more in the potential role of this analytic method in identifying potentially effective therapies in complex diseases. In light of this ore generic interest, I would ask the authors to address the following specific comments.

Specific comments

1. Sepsis is a poorly characterized clinical syndrome; how consistently is it defined across your study cohorts?

2. You speak in the first sentence of sepsis as being characterized by elevated cytokine levels. This is simplistic and inaccurate, given that the majority of genese impacte in sepsis are down-regulated. Please try to be more biologically precise in your descriptions.

3. Your hypothesis would suggest a similar role for variants in gp130: can you analyze this as a complementary perspective on youor work?

4. Figure 1 is confusing: why is a distinction between cis and trans signaling relevant to your hypothesis?

5. Your variously speak of 28 and 30 day survival throughout the paper; please clarify.

6. Why do you differentiate lower respiratory tract infection and pneumonia: to clinicians these are the same entity, or at least there is significant overlap.

7. The stamen (PDF p 16) that "Conceptually our genetic instrument is similar to the action of anti-IL6R monoclonal antibodies (e.g. tocilizumab) that lead to complete inhibition of IL6R signalling by blocking both IL6 classical and trans signaling" is a bit of an over-reach. Monoclonal antibodies do not necessarily result in complete inhibition andthe genomic region you study is not limited to IL-6R.

8. The effect for endocarditis seems odd: can you elaborate?

9. In Table 1, the effects in sepsis are modest; the confidence intervals are wide, and much less impressive that what is seen for COVID.

10. The use of CRP as a marker of IL-6 bioactivity is potentially complicated because it is an acute phase reactant, identified through its role in pneumococcal infection, and so potentially impacted by other biologic influences.

11. There are a number of grammatical errors or awkward formulations throughout. A personal peeve is the use of the word "interrogate" to describe a scientific experiment. One interrogates people, and may use inappropriate methods of suasion to gain a response; science tests hypotheses, and hopefully minimizes the biassess associated with this.

Reviewer #3: The present study investigated whether the blockade of IL6R improves the outcome of sepsis. IL-6 is a pleiotropic cytokine, and the related SNPs of concern are recognized as candidate markers to select genetically high-risk patients in ICU. The study is profoundly designed, and the data are also robust. Investigating whether this SNP is meaningful among abundant candidates in this field is essential. It might be a good idea to discuss the rs2228145 SNP to deepen readers' understanding of the genetic marker.

Reviewer #4: The current ms aimed at the potential value of IL-6/IL6R blockade in intensive care settings. This treatment might improve survival similar to critically ill patients suffering from COVID-19.

Appropriate statistical methods were applied and the diagnosis of patients included were vbased on ICD-10 classification. Different aspects guiding sensitivity were thoroughly included and interpretation was done with the necessary caution and knowledge.

Schematics and data presentation are also appropriate.

Based on the either detrimental or beneficial effects by anti-IL-6 treatment (due to IL6R signaling), the current data sets are valid to motivy clinicians for a clinical study using IL-6/IL6R blockade in ICU patients independently of COVID-19.

Since critically ill patients also suffer from a number of illnesses before manifestation of sepsis, functional SNPs may play a significant role as well.

It may constitute a valuable aspect to include th recently published work by Zhang et al. 2022 (doi: 10.3389/fimmu.2022.860703) in the discussion. Finally, this may lead to an improved concept for clinical studies

[LINK]

---

## [Decision Letter · Decision Letter 2]

9 Jan 2023

Dear Dr. Hamilton,

Thank you very much for re-submitting your manuscript "Therapeutic potential of IL6R blockade for the treatment of sepsis and sepsis-related death: a Mendelian randomisation study’" (PMEDICINE-D-22-02409R2) for review by PLOS Medicine.

I have discussed the paper with my colleagues and the academic editor and it was also seen again by two reviewers. I am pleased to say that provided the remaining editorial and production issues are dealt with we are planning to accept the paper for publication in the journal.

[LINK]

We look forward to receiving the revised manuscript by Jan 16 2023 11:59PM.   

Sincerely,

Callam Davidson, 

Senior Editor 

PLOS Medicine

plosmedicine.org

Requests from Editors:

Data Availability Statement: ‘Access to the full summary statistics for the sepsis GWAS performed by the GaINS and GenOSept committee is by application to the relevant committee. This research was performed under UK Biobank application 56243. Individual access to UK Biobank can be arranged via the UK Biobank website.’ Please include relevant contact details (websites or email addresses) for data enquiries.

Please include paragraph numbers as well as section headings in your STROBE-MR checklist. 

Please ensure citations precede punctuation throughout. 

The Funding, Competing Interests, and Data Availability sections can all be removed from the main text – this information is captured via the Submission Form and will be published as metadata alongside the article after acceptance.

Comments from Reviewers:

Reviewer #1: Minor comments:

1. The authors have added the following small detail to the methods and state that alternative meta-analytic approaches like MR-Egger and weighted median can test all three instrumental variable assumptions: 

"We performed three broad types of sensitivity analysis. Firstly, we attempted to test the assumptions of MR (relevance, exclusion restriction, and indepenece) using alternative meta-analytic approaches (MR-Egger and weighted median approaches)."

Importantly, MR-Egger and weighted median approaches only test for violations of the exclusion restriction assumption. They cannot verify that the exclusion restriction assumption holds, but only detect violations. Moreover, these methods do not test the relevance (to be tested with the F-statistic) or independence (untestable in summary-level setting) assumption. Please correct.

2. Please correct the following typos indepenece and pleotropic

Reviewer #2: Thank you for your thoughtful response to my comments, and for pushing back when you deem appropriate. Quite apart from the potential impact of providing further support for a trial of IL-6ra in sepsis, your work introduces and helps to clarify the role of an appealing approach to address heterogeneity in complex diseases.

[LINK]

---

## [Editor Report · Decision Letter 3]

13 Jan 2023

Dear Dr Hamilton, 

On behalf of my colleagues and the Academic Editor, Dr Cosetta Minelli, I am pleased to inform you that we have agreed to publish your manuscript "Therapeutic potential of IL6R blockade for the treatment of sepsis and sepsis-related death: a Mendelian randomisation study’" (PMEDICINE-D-22-02409R3) in PLOS Medicine.

PRESS

Sincerely, 

Callam Davidson 

Associate Editor 

PLOS Medicine